# Knee Kinetics and Kinematics of Young Asymptomatic Participants during Single-Leg Weight-Bearing Tasks: Task and Sex Comparison of a Cross-Sectional Study

**DOI:** 10.3390/ijerph19095590

**Published:** 2022-05-04

**Authors:** Gustavo Luís Bellizzi, Tenysson Will-Lemos, Renan Alves Resende, Ana Cristina Corrêa Cervi, Paulo Roberto Pereira Santiago, César Fernández-de-las-Peñas, Débora Bevilaqua-Grossi, Lidiane Lima Florencio

**Affiliations:** 1Department of Health Sciences, Ribeirão Preto Medical School, University of São Paulo, Ribeirão Preto 14049-900, Brazil; glbellizzi.fisio@hotmail.com (G.L.B.); tenysson@fmrp.usp.br (T.W.-L.); aninhacris_@hotmail.com (A.C.C.C.); deborabg@fmrp.usp.br (D.B.-G.); 2Department of Physical Therapy, Universidade Federal de Minas Gerais (UFMG), Belo Horizonte 31270-901, Brazil; renan.aresende@gmail.com; 3School of Physical Education and Sport of Ribeirão Preto, University of São Paulo, Ribeirão Preto 14040-907, Brazil; paulosantiago@usp.br; 4Department of Physical Therapy, Occupational Therapy, Rehabilitation and Physical Medicine, Universidad Rey Juan Carlos, 28922 Alcorcón, Spain; cesar.fernandez@urjc.es

**Keywords:** kinetic, inverse dynamics, knee joint, kinematic, exercise, functional assessment, biomechanics

## Abstract

This cross-sectional study aimed to describe and compare kinetic and kinematic variables of the knee joint during stair descent, single-leg step down, and single-leg squat tasks. It also aimed to investigate potential sex difference during the tasks. Thirty young asymptomatic individuals (15 males, 15 females) were assessed during the performance of single-leg weight-bearing tasks. The kinetic and kinematic data from the knee were evaluated at the peak knee moment and at peak knee flexion. Single-leg squat presented a higher peak knee moment (2.37 Nm/kg) and the greatest knee moment (1.91 Nm/kg) at knee peak angle in the frontal plane, but the lowest knee flexion (67°) than the other two tasks (*p* < 0.05). Additionally, the single-leg step down task presented a higher varus knee angle (5.70°) when compared to stair descent (3.71°) (*p* < 0.001). No substantial sex difference could be observed. In conclusion, in asymptomatic young individuals, single-leg squats presented the greatest demand in the frontal and sagittal planes. Single-leg step down demanded a greater angular displacement than stair descent in the frontal plane. We did not identify a significant difference among the sex and studied variables.

## 1. Introduction

The functional assessment of the knee joint has received a great deal of attention in recent years in clinical and scientific fields [1]. Specific focus on the knee could be attributed to the high incidence of pain [2,3] and injuries in this joint [4,5,6,7,8].

Tasks such as bipedal and single-leg squats, stair descents, and landings compose the functional assessments for knee joints [1]. Single-leg tasks are also commonly used during the rehabilitation process, as a clinical assessment tool, and as rehabilitation exercises [1,9,10,11]. However, different single-leg tasks certainly generate distinct biomechanical demands, as observed for variations of jump landings [12], squats [13,14], and functional exercises [15]. Current evidence indicates that kinematics and kinetic aspects are not risk factors for knee injuries [16,17,18], except when they are intrinsically related to the task, such as running-related injuries [19]. Still, these biomechanical differences may be considered when planning the exercise protocols. Depending on the rehabilitation’s objective or phase (early or advanced), some patients might have therapeutic restrictions considering the articular range of motion or benefit from a progressive demand according to the rehabilitation’s objective or phase (early or advanced).

Unipedal tasks require less knee joint angulation than bipedal tasks [12,20,21,22,23,24]. However, depending on the task type, greater [20,23,24] or smaller joint moments have been observed [12,21]. When comparing single-leg tasks, single-leg squats present a greater angle of knee flexion and smaller angles of knee abduction and abductor moments than single-leg landings of bipedal horizontal jumps [21]. Single-leg squats also present a greater angle of flexion and knee abduction when compared to single-leg step downs [25].

In addition to task type, another factor that must be considered is the potential difference between sex. Cronström et al. [26] demonstrated in a meta-analysis that women have higher knee abduction peaks during stair descents, jumping, and unipedal landings; however, there was no evidence of differences between sex in single-leg squatting. In addition, women had lower knee flexion and medial rotation angles and lower peaks of knee extensor moment and ground reaction forces during stair descent tasks [27,28]. Recognizing a sex-related biomechanics’ strategy to perform the unipedal tasks may help interpret the results obtained during the functional assessment and customize orientations to perform the exercises.

Therefore, we can hypothesize that the type of task and sex can influence joint moments during single-leg tasks. No published studies have compared the biomechanical demands or, especially, kinetic data for stair descent, single-leg squat, and single-leg step down tasks considering potential differences between sex. Understanding how these tasks influence the knee joint moment in both sexes complements the knowledge about the biomechanical demand to which individuals are often exposed. It may also guide decision-making, as increasing biomechanical demand progressions are recommended in the knee rehabilitation process [15].

The objective of this study was to describe and compare kinetic and kinematic variables of the knee joint during stair descent, single-leg step down, and single-leg squat tasks in young asymptomatic participants. The secondary objective was to investigate if there was a difference between sex in the performance of these tasks. In addition, it aimed to describe and compare the kinetic and kinematic variables of the adjacent hip and ankle joints at the peak joint moment of the knee.

## 2. Materials and Methods

### 2.1. Participants

Individuals between 18 and 35 year of age were recruited from the local community to participate in this cross-sectional study. The inclusion criteria were participants with no complaints of pain in their lower limbs and who were classified as “active” or “irregularly active” according to the International Physical Activity Questionnaire (IPAQ) [29]. The exclusion criteria were reports of pain in the last year, a history of lesions or surgeries in the lower limbs, neurological disorders, and women currently in or one week prior to their menstrual periods. All participants signed a free and informed consent form approved by the local ethics committee (protocol no. 14961/2014).

A total of 33 individuals were recruited, of which 3 were excluded for being classified as “very active” (*n* = 2) or sedentary (*n* = 1). The final sample included 30 individuals (15 males and 15 females), and their characteristics are described in Table 1. There were significant differences between sexes in the height and weight of the participants (*p* < 0.05).

### 2.2. Procedures

The Vicon motion analysis system (Centennial, CO, USA) was used for biomechanical evaluations. It was composed of 8 cameras (MX-T-40S) with 4 megapixels of spatial resolution and a sampling frequency of 250 Hz, synchronized with two BERTEC (Columbus, OH, USA) and AMTI Accugait (Waterloo, MA, USA) force platforms, both with sampling frequencies of 2000 Hz.

A total of 42 retroreflective markers (20 mm in diameter) were positioned on the following anatomical landmarks: spinous process of the 10th thoracic vertebra; jugular notch; acromion; posterior superior iliac spine; anterior superior iliac spine; prominence of the greater trochanter, the external surface of the femur; lateral condyle femur; medial condyle femur; head of the fibula; prominence of the tibial tuberosity; distal apex of the medial malleolus; distal apex of lateral malleolus; calcaneus base; the midpoint of Achilles’ tendon, dorsal aspect of the second metatarsal head; dorsal aspect of the second metatarsal base; dorsomedial aspect of the first metatarsal head; dorsomedial aspect of the first metatarsal base; dorsolateral aspect of the fifth metatarsal base; dorsolateral aspect of the fifth metatarsal head; medial aspect of the head of the proximal phalanx of the hallux; most medial apex of the tuberosity of the navicular [30,31,32,33]. These data were captured in the orthostatic position with the individual in an anatomical position to align with the global coordinate system and identify the articular axes.

Participants wore workout clothes (tight shorts, and women also wore a top) and performed the tasks without shoes. They had no standard rest period before familiarization. However, the time to collect participant data, position the retroflective marks, and the system’s calibration was a sufficient interval to rest. Then, participants performed 4 repetitions of each test (stair descent, single-leg step down, and single-leg squat tasks) to familiarize themselves with the tasks. Then, they performed 3 valid tests with their dominant lower limb as the stance limb, which was identified by the preference to kick a ball. There was an interval of 60 s between each valid test. All participants performed the same test order, starting with the stair descent, followed by the single-leg step down, and finalized the assessment with the single-leg squat tasks.

Participants started from a standing bipedal position for the stair descent task with their arms positioned at the sides of their bodies on the highest stair step. They descended the steps using their typical gait pattern, starting with their non-dominant lower limb [34,35]. The steps had a fixed height of 20 cm, and the force platform (AMTI) was positioned on the step closest to the ground (Figure 1A). For the single-leg step down task, the participants stood in a bipedal stance with the base of their feet shoulder-width apart and arms crossed in front of their chests on a step 20 cm in height, gently and slowly touched the ground with the heel of their non-dominant lower limb after a voice command, and then returned to their initial position (Figure 1B). For the single-leg squat, the participants stood in a bipedal stance with their arms crossed in front of their chests on the force platform (BERTEC) and, at a voice command, lifted their non-dominant limbs off of the platform, performing a 4 s squat (2 s to perform the squat and 2 s to return to their initial position), without touching any surface with their non-dominant lower limb (Figure 1C).

### 2.3. Data Processing

The retroreflective markers were three-dimensionally reconstructed using VICON NEXUS 1.8.5 software, and the data were processed using Visual3D software (version 6.01.22, C-Motion Inc., Germantown, NY, USA)). The kinematic data were filtered using a fifth-order Butterworth low-pass filter with 8 Hz cut-off frequencies. The articular center of the ankle was determined by the midpoint between the lateral malleolus, and the medial malleolus, the center of the knee joint was represented by the midpoint between the lateral and medial femoral epicondyles. The center of the hip was represented by the markers on the right and left anterosuperior iliac spine and the midpoint between the two posterosuperior iliac spines [23,24,36]. The internal joint moments of the ankle, knee and hip in the frontal and sagittal planes were calculated using inverse dynamics and normalized by the weight and height of the participants.

The joint coordination system was used to analyze the hip, knee, and ankle angles [37], adopting an XYZ sequence. First, a Cartesian coordinate system was established for each of the two adjacent body segments, defined based on bony landmarks. The common origin of both systems defined the neutral position, and it was the reference for the linear translation. Secondly, the joint coordination system was established based on the two Cartesian coordinate systems. Two of the joint coordination system axes were body fixed, and one was not fixed. Finally, the joint motion, including three rotational and three translational components, was defined based on the joint coordination system [38]. Local reference systems of the thigh and leg were built to calculate the knee joint angles. Markers on the greater trochanter of the femur and the lateral and medial epicondyles of the femur were used to build the thigh segment bases. The markers on the head of the fibula, lateral malleolus, and medial malleolus were used to build the leg segment bases. The hip segment base was built with the markers on the right and left anterosuperior iliac spine and the midpoint between the markers of the two posterosuperior iliac spines. The angles were calculated using this base associated with the thigh base. The foot segment bases were used for the ankle, which consisted of the calcaneal markers and the heads of the first and fifth metatarsals. The angles were calculated using this base associated with the leg base. The definitions of the joint coordinate system recommended by the International Society of Biomechanics [38] were used.

The X-axis should represent the flexion (+) and extension (−), and the Y-axis should represent the adduction (−) and abduction (+) [38]. For a better clinical understanding of the results, the angles of the knee in the frontal plane were defined by valgus [adduction (−)], and varus (abduction (+)), and the sagittal angles of the ankle were defined by dorsiflexion (flexion (+)) and plantar flexion [extension (−)] (Figure 2).

The stair descent task was limited to the initial contact of the tested lower limb with the ground (using the vertical component of the ground reaction force) to the removal of this limb from the force platform. The single-leg step down task was limited to the removal of the contralateral limb from the ground and followed by a squat at maximum flexion until the return of this same limb to the ground. Finally, the single-leg squat task was limited to the removal of the contralateral limb from the ground, followed by a squat at maximum flexion until the return of this same limb to the ground.

Once the appropriate phases of each task had been defined, the frames of the an-gular and joint moment peaks of the knee in the frontal and sagittal planes were identified, representing the position and load of greatest demand for this joint. Knee angle and joint moment were calculated for each frame of interest. In addition, the angle and moment of the hip and ankle were identified from the peak knee joint moment frame in their respective plane.

### 2.4. Statistical Analysis

SPSS 21.0 software (IBM Corporation, Armonk, NY) was used for statistical analysis of results at a significance level of 0.05. Sample characterization variables were compared with the Student’s *t*-test for independent samples and described as means and standard deviations (SD) or with the Chi square test and described by their proportions.

As kinetic and kinematic data violated the assumption of normal distributions, even after logarithmic transformation, non-parametric tests were used to compare groups. According to them, data were described as median and interquartile ranges. The Friedman test was used to compare kinetic and kinematic variables in the 3 tasks, and a post hoc Wilcoxon test was used with Bonferroni correction for multiple comparisons. The Mann–Whitney test was used to compare sex in each of the tasks.

## 3. Results

### 3.1. Knee Kinetics and Kinematics during Single-Leg Weight-Bearing Tasks

All data about the knee kinetics and kinematics are represented in Figure 3, and their respective numerical medians and interquartile range can be consulted in Appendix A.

In the frontal plane, single-leg squats showed a higher peak of adductor moment for the knee (−2.37 Nm/kg) compared to stair descents (−0.88 Nm/kg; *p* < 0.001) and single-leg step downs (−1.00 Nm/kg, *p* = 0.002) (Figure 3A). The varus angulation of the knee at its peak joint moment in the frontal plane was significantly lower in the stair descent tasks (+1.44°) when compared to single-leg squats (+2.09°, *p* = 0.01) and single-leg step downs (+3.39°, *p* = 0.009) (Figure 3B).

Considering the angular peak of the knee identified in each task, there is a greater varus knee angle in the single-leg step down (+5.70°) than in the stair descent (+3.71o) (*p* < 0.001) (Figure 3D). At the peak angular peak of the knee, a significant joint moment difference between all three tasks was identified in the frontal plane (X² = 38,600 (2), *p* < 0.001), with decreasing order of the adductor joint moments for single-leg squats (−1.91 Nm/kg), single-leg step downs (−0.87 Nm/kg), and stair descents (−0.15 Nm/kg) (Figure 3C).

In the sagittal plane, no task differences were identified at the peak joint moment of the knee (X² = 4867 (2), *p* = 0.088) (Figure 3E). However, at this peak joint moment, single-leg squats showed lower angulation in flexion (56.6°) when compared to single-leg step downs (67.9°) (*p* = 0.001). Knee angulation at stair descent (54.1°) was also significantly lower than that of single-leg step down (68°) (*p* = 0.009) at the peak joint moment of the knee (Figure 3F).

The single-leg squat task demanded the lowest peak of knee flexion (+67.14°) compared to that of stair descent (+86.7°, *p* < 0.001) and single-leg step down (+75.4°, *p* = 0.001) (Figure 3H). However, it was the task that presented the greatest joint moment in this reference frame of angular knee peak (−2.03 Nm/kg), being identified as a con-centric flexor moment when compared to stair descent which presented an eccentric extensor moment (+0.09 Nm/kg, *p* < 0.001) (Figure 3G). The same pattern was observed between stair descent and single-leg step down (−1.40 Nm/kg, *p* < 0.001), but the latter did not differ significantly from single-leg squats.

### 3.2. Sex Differences of Knee Kinetics and Kinematics

There was no evidence of sex differences for most kinetic and kinematic variables of the knee. Only single-leg squats showed that males (+3.18°) presented a greater varus angle of the knee than females (+0.08°) (U = 64,000, *p* = 0.045) at the peak joint moment of the knee in the frontal plane (Figure 3B) (Appendix A).

### 3.3. Hip and Ankle Kinetics and Kinematics at Peak Knee Joint Moment

All kinetic and kinematic from hip and ankle during the peak knee joint moment are represented in Figure 4. Additionally, their medians and interquartile range can be consulted in Appendix A.

Hip kinetics at peak knee joint moment differ among tasks in both planes. In the frontal plane, the hip had a greater abductor moment in single-leg squats (+2.75 Nm/kg) than in stair descents (+0.98 Nm/kg, *p* < 0.001) or single-leg step downs (+1.29 Nm/kg, *p* = 0.001). In sagittal plane, the hip flexor moment was greater for stair descents (+2.66 Nm/kg) than for single-leg squats (+0.61 Nm/kg, *p* = 0.001) or single-leg step downs (+0.28 Nm/kg, *p* = 0.001). No differences were found in ankle joint moments at the peak knee joint moment in the frontal and sagittal planes (Figure 4).

Differences among tasks could also be observed for the kinematic variables of the adjacent joints at the peak knee joint moment. In the frontal plane, greater hip adduction was observed for single-leg step downs (−9.98°, *p* < 0.001) and single-leg squats (−5.33°, *p* = 0.006) than for stair descents (−0.86°) without differing from one another. Lower ankle adduction was seen for single-leg step downs (−5.07°, *p* < 0.001) and single-leg squats (−5.60°, *p* < 0.001) compared to stair descents (−12.74°) without differing from one another. In the sagittal plane, a greater hip flexion was seen for single-leg squats (+39.7°, *p* < 0.001) and single-leg step downs (+45.4°, *p* < 0.001) when compared to stair descents (+5.0°), without differing from one another. For the ankle, single-leg step downs (+10.11, *p* = 0.014) presented greater dorsiflexion compared to stair descent (+2.9°) (Figure 4).

Sex differences were observed in adjacent joints at the peak of the knee joint moment. In the frontal plane, males presented hip abduction (+2.41), while females presented hip adduction (−2.27) in stair descent (*p* = 0.003). In the sagittal plane, females had a higher plantar flexor moment (−0.59) in single-leg squats than males (−0.16) (*p* = 0.04) (Figure 4).

## 4. Discussion

The current results indicated that single-leg squats presented greater knee joint moments in the frontal plane. Although all three tasks showed no differences in the peak joint moment in the sagittal plane, single-leg squats and single-leg step downs had greater knee joint moments when the knee was at maximum flexion than stair descents. However, single-leg squats presented the lowest knee flexion among the three tasks. In the frontal plane, stair descents presented the lowest angulation differing significantly only from single-leg step downs. All the differences mentioned were observed only when the total sample was considered, and no substantial differences between the sexes were observed.

No reports were found in the literature comparing articular moments between the tasks evaluated in this study. However, the median knee joint moment in the single-leg squat was higher than the mean reported in previous studies on peak joint moments in sagittal planes for males [20], peak frontal moments for both sexes [21], and knee joint moments in angular peaks in both planes and sexes [13,14]. These differences may be due to several factors, including: the lack of standardization for the position of the upper and contralateral lower limbs when performing single-leg squats; the use of different events to determine the start and end of tasks; and the use of different ways to calculate and describe the joint moment.

It was possible to verify that single-leg squats were also the task in which there was a greater hip joint moment at the peak knee joint moment in the frontal plane. However, this was not true for the ankle joint, in which no differences were identified at the articular moment among tasks. In contrast, the positioning of adjacent joints suggests different strategies even when there was no difference in the peak knee joint moment, with stair descents presenting the lowest angles of flexion and adduction of the hip and greater adduction and less ankle dorsiflexion in this reference frame.

As for kinematic variables of the knee joint, a lower flexion peak at single-leg squats compared to the other two tasks corroborated the findings by Lewis et al. [25]. They also observed a lower flexion peak in single-leg squats compared to single-leg step downs. It is essential to highlight that the height of the step can influence these results, since differences between single-leg squats and single-leg step downs were reported using steps with heights of 20 and 16 cm, but not when single-leg step downs were performed at a step height of 24 cm [25]. The step height to perform the single-leg step down may determine distinct trunk and lower limb strategies since the individual has to touch the ground with the heel of their non-dominant lower limb [25]. Specifically, considering the knee, the greater the step height, a greater knee flexion would be necessary.

The hypothesis of differences between sex and kinetic and kinematic variables of the knee at single-leg tasks was not confirmed in most of the analyses in this study. The only difference found was in single-leg squats. We identified a greater varus angulation for males at the peak knee joint moment, the same observation reported by Zawadka [39]. Although Khuu and Lewis [14] also reported sex differences in single-leg squats, the differences were demonstrated only by a smaller flexor moment at the angular knee peak for females during single-leg squats. A meta-analysis by Cronstöm et al. [26] showed no sex differences in knee angulation in the frontal plane during single-leg squats. However, we cannot compare our results to theirs since none of the reviewed studies verified angulation at the peak knee joint moment in the frontal or sagittal planes.

No sex differences were identified for the kinetic and kinematic variables of the knee during stair descents or single-leg step downs. These findings corroborate the results reported by Baldon et al. [27], who did not observe kinematic differences in the sagittal plane during stair descents. However, they contradict the higher knee abduction results for single-leg step downs [22] and the lower extensor moments for stair descents [28] previously reported for young healthy females compared to males. Again, these differences can be explained by the height of the step since a similar height of 20.5 cm was used in the study [27] corroborates the present study and higher steps (23 cm and 30 cm) were used in studies where results differed from the present study [22,28]. The influence of step height had already been highlighted by potential knee angulation differences between sex in the frontal plane for landing tasks [26].

The difference in height between the groups can be seen as a potentially confusing factor [28] since it directly influences the distance from the center of mass in relation to the point of contact. However, the normalization of data by height and weight in this study eliminates this influence in the joint moment. A limitation of the present study is the impossibility of verifying the interaction between the two factors (tasks and sex) due to the distribution of residual data, which could show different patterns in performing the tasks. Maybe distinct distributions could be used or different statistical methods with larger sample sizes applied. It would also be desirable to perform a multivariate analysis. As we did not perform a priori sample size estimation, we might have been underpowered to detect some subtle differences between tasks or sexes. In addition, future studies can investigate differences in the transverse plane and use different step heights to complement our results and thus improve the understanding of the demands required for each task. Another perspective for future studies is the inclusion of sedentary and very active participants, considering the activity level as a factor that could be associated with the kinetic and kinematic parameters.

Strengths of this study include comparing three single-leg weight-bearing tasks, which are highly used in practice but, to the best of our knowledge, have never been biomechanically compared before. Understanding the similarities and differences among the three assessed tasks and between sexes may guide some practical and/or clinical decisions. For example, if the aim is to expose a young and asymptomatic individual to a single-leg weight-bearing tasks program, our results would guide to adequate the exercises to the objectives. Our results suggest that the progressive implementation of single leg-tasks in the rehabilitation program may begin with stair descent followed by a single-leg step down and finally the implementation of a single-leg squat when considering the articular moment. Moreover, in terms of range of motion in knee frontal plane, clinicians should be aware that the single-leg step down is the one that requires the greater range. So, if the goal is to stimulate greater knee mobility in the frontal plane, the single-leg step down would be preferred among the three assessed herein; however, if there is any clinical precaution to such mobility, this task could be avoided at this timepoint. Finally, as we did not observe any differences in the knee peak angle in the sagittal plane, all the three tasks could be equally chosen.

Several factors can contribute to altered kinetics and kinematics during single-leg weight-bearing tasks, such as active or passive stiffness, altered neuromuscular control, or muscle strength [40,41,42,43]. It should be highlighted that, currently, there is no evidence that altered and/or distinct kinetic and kinematic parameters are risk factors for knee-related injuries or painful conditions [16,17,18]. Nevertheless, altered biomechanics can be observed in patients with anterior cruciate ligament repair, knee osteoarthritis, and patellofemoral pain [44,45,46]. However, as we did not assess any pathology or painful knee condition, the practical implications mentioned before may not be the same for these cases.

## 5. Conclusions

The study results show that in asymptomatic young individuals, single-leg squat present the greatest demand compared to single-leg step down and stair descent tasks in the frontal and sagittal planes. Single-leg step down seems to demand greater angular displacement than stair descent in the frontal plane. These results could help choose which single-leg task would fit better within a specific clinical purpose. For example, the single-leg step down could be used if the goal is to expose the individual to greater articular demand. Finally, we did not identify a significant difference between the sex and the studied variables.

## Figures and Tables

**Figure 1 ijerph-19-05590-f001:**
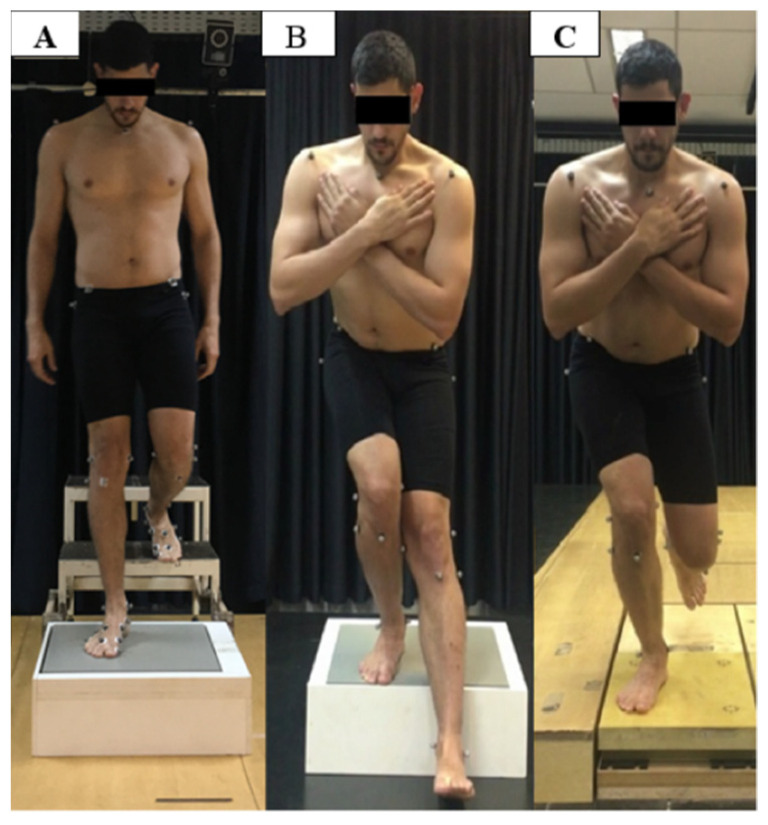
Single-leg support tasks: step descent (**A**), single-leg step down (**B**), and single-leg squat (**C**).

**Figure 2 ijerph-19-05590-f002:**
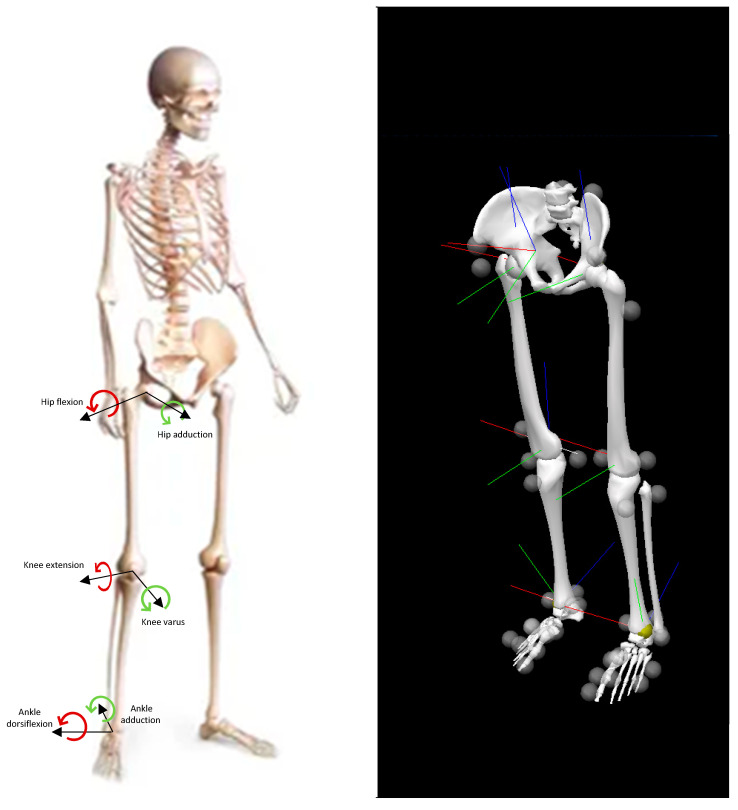
Graphic representation of the nomenclature adopted to describe the lower limb movements and angles and their respective signal in the frontal and sagittal planes.

**Figure 3 ijerph-19-05590-f003:**
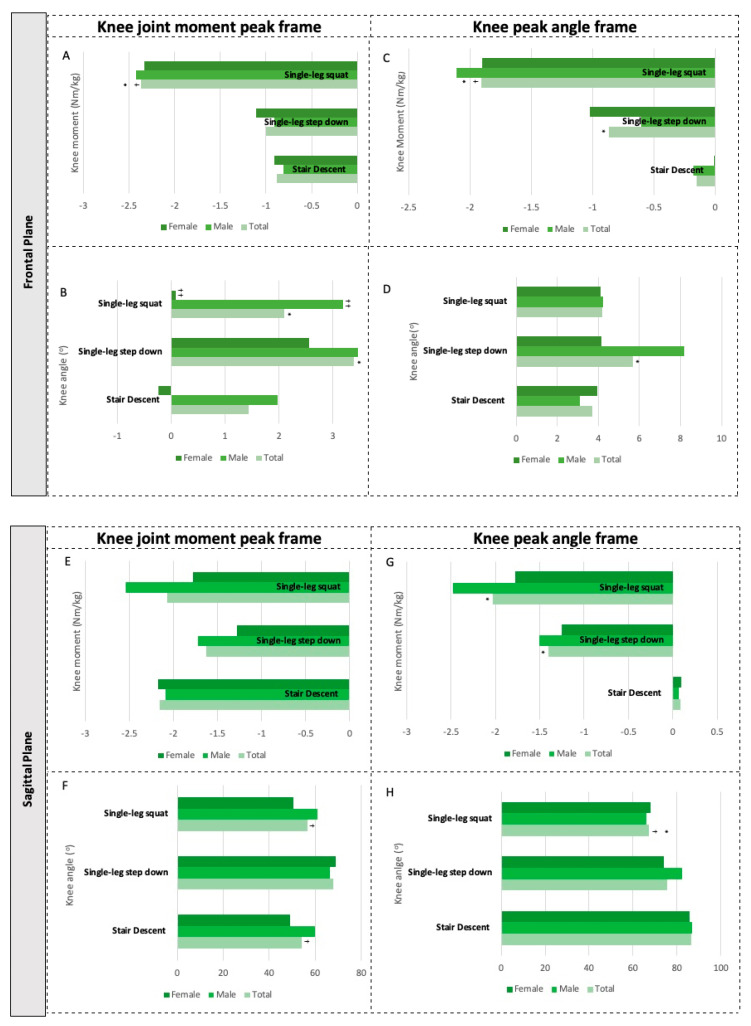
Knee kinetics and kinematics during single-leg weight-bearing tasks of asymptomatic young people (*n* = 30) and stratified by sex (15 females, 15 males): median in the moment of occurrence of the articular peak moment (**A**,**B**,**E**,**F**) and the angular peak of the knee (**C**,**D**,**G**,**H**) in both the frontal and sagittal plane during stair descent, single-leg step down and single-leg squat. Subfigures are displayed according to the plane and the frame: the knee moments and angle at the frontal plane from the knee joint knee peak frame (**A**,**B**) and from the knee peak angle frame (**C**,**D**); the knee moments and angle at the sagittal plane from the knee joint knee peak frame (**E**,**F**) and from the knee peak angle frame (**G**,**H**). Note: Symbols represent in the frontal plane abduction/varus (+) and adduction/valgus (−) of the knee, and in the sagittal plane, flexion (+) extension (−) of the knee. * Different stair descent (*p* < 0.05). † Different from single-leg step down (*p* < 0.05). †† Sex difference (*p* < 0.05).

**Figure 4 ijerph-19-05590-f004:**
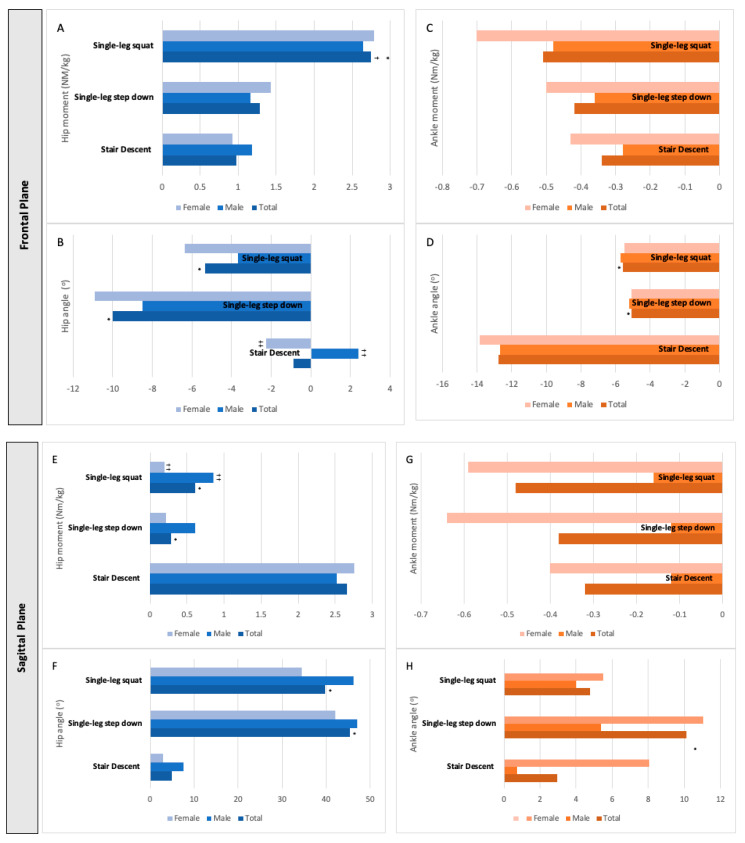
Adjacent joints: kinetic (**A**,**C**,**E**,**G**) and kinematic (**B**,**D**,**F**,**H**) data of the hip (blue bars) and ankle (orange bars) joints at the moment of peak knee joint moment of the total sample (*n* = 30) and stratified by sex (male, *n* = 15; female, *n* = 15). Subfigures are displayed according to the plane: (**A**,**B**) describe the hip moment and angle at the frontal plane, (**C**,**D**) describe the ankle moment and angle at the frontal plane; (**E**,**F**) describe the hip moment and angle in the sagittal plane, (**G**,**H**) describe the ankle moment and angle in the sagittal plane. Note: Symbols represent in the frontal plane abduction (+) and adduction (−) for hip and ankle, and in the sagittal plane, hip flexion (+), extension (−) or dorsiflexion (+), and plantar flexion (−) of the ankle. * Different stair descent (*p* < 0.05). † Different from single-leg step down (*p* < 0.05). †† Sex difference (*p* < 0.05).

**Table 1 ijerph-19-05590-t001:** Mean and standard deviation of sample characteristics, proportions of lower limb dominance and International Physical Activity Questionnaire (IPAQ) classification.

	Total Sample (*n* = 30)	Males (*n* = 15)	Females (*n* = 15)	*p* Value of Sex Comparison
Age (years old)	21.93 (3.00)	22.80 (3.28)	21.07 (2.52)	0.116
Height (m)	1.70 (0.08)	1.76 (0.06)	1.66 (0.06)	<0.001
Weight (kg)	71.80 (11.47)	79.47 (6.06)	64.13 (10.47)	<0.001
Lower limb dominance				
Right	26 (87%)	13 (87%)	13 (87%)	1.000
Left	4 (13%)	2 (13%)	2 (13%)	
IPAQ classification				
Active	3 (10%)	3 (20%)	1 (7%)	0.28
Irregularly active	27 (90%)	12(80%)	14 (93%)	

## Data Availability

The data presented in this study are available upon request from the corresponding author. The data are not publicly available due to ethical issues.

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
