# Peer review of "Knee Kinetics and Kinematics of Young Asymptomatic Participants during Single-Leg Weight-Bearing Tasks: Task and Sex Comparison of a Cross-Sectional Study"

_ijerph, 2022, doi:10.3390/ijerph19095590_

Round 1

Reviewer 1 Report

In the present study, the authors pretend to biomechanically analyse 3 functional knee tests currently used. It is correctly designed and appropriately analyzed.

The English should be revised. Please refine expressions such as bi and unipedal.

Introduction

The introduction is relatively well written, however, the problem could be better identified. The same for the rationale for considering sex comparisons.

Methods

Refer if the testing order was randomized, and add the ankle marker placement used.

Results

Are presented satisfactorily, although if the tables are formatted it would improve their readability.

Line 172 Revise this value (“compared to stair descents (0.88 Nm/kg;” ) as it is not the same in table 2

Discussion

Please discuss the step's high eventual influence on some of the results obtained.

Conclusion

Are presented satisfactorily. However, I believe the paper would beneficiate from highlighting the study's relevance to clinical practice.

Urkund report: About 7% of this document consists of text similar to text found in 64 sources. The largest marking is 30 words long and is 100% similar to its primary source.

Author Response

We would like to thank you for your time spent and comments. We believe that the suggestions and queries from the review helped to improve the quality of our research report. We have followed the suggestions and highlighted all changes in yellow throughout the text. We have included a point-by-point cover letter.

In the present study, the authors pretend to biomechanically analyse 3 functional knee tests currently used. It is correctly designed and appropriately analyzed.

Response: We appreciate your positive comment.

The English should be revised. Please refine expressions such as bi and unipedal.

Response: Thank you for the suggestion. The English was revised, and we changed the expressions “bi and unipodal” to “bi and unipedal” as suggested.

Introduction

The introduction is relatively well written, however, the problem could be better identified. The same for the rationale for considering sex comparisons.

Response: We appreciate your comment. We revised the Introduction section and added the following sentences to highlight the problem regarding the tasks and sex comparisons.

Page 1, line 36 to page 2, line 47:     Tasks such as bipedal and single-leg squats, stair descents, and landings compose the functional assessments for knee joints [1]. Single-leg tasks are also commonly used during the rehabilitation process, as a clinical assessment tool, and as rehabilitation exercises [1,9-11]. However, different single-leg tasks certainly generate distinct biomechanical demands, as observed for variations of jump landings [12], squats [13, 14], and functional exercises [15]. Current evidence indicates that kinematics and kinetic aspects are not risk factors for knee injuries [16-18], except when they are intrinsically related to the task, such as running-related injuries [19]. Still, these biomechanical differences may be considered when planning the exercise protocols. Depending on the rehabilitation's objective or phase (early or advanced), some patients might have therapeutic restrictions considering the articular range of motion or benefit from a progressive demand according to the rehabilitation's objective or phase (early or advanced).”

Page 2, lines 54-62: “In addition to task type, another factor that must be considered is the potential difference between sex. Cronström et al. [26] demonstrated in a meta-analysis that women have higher knee abduction peaks during stair descents, jumping, and unipedal landings; however, there was no evidence of differences between sex in single-leg squatting. In addition, women had lower knee flexion and medial rotation angles and lower peaks of knee extensor moment and ground reaction forces during stair descent tasks [27, 28]. Recognizing a sex-related biomechanics' strategy to perform the unipedal tasks may help interpret the results obtained during the functional assessment and customize orientations to perform the exercises.

Methods

Refer if the testing order was randomized, and add the ankle marker placement used.

Response:  Thank you for your suggestion. We described all the anatomical landmarks (Page 3, lines 104-114), clarified which ones were used for the foot base (Page 4, lines 170 to page 5 line 172) and added the testing order was not randomized (page 3, lines 124-126):

Page 3, lines 104-114A total of 42 retroreflective markers (20 mm in diameter) were positioned on the following anatomical landmarks: spinous process of the 10th thoracic vertebra; jugular notch; acromion; posterior superior iliac spine; anterior superior iliac spine; prominence of the greater trochanter, the external surface of the femur; lateral condyle femur; medial condyle femur; head of the fibula; prominence of the tibial tuberosity; distal apex of the medial malleolus; distal apex of lateral malleolus; calcaneus base; the midpoint of Achilles' tendon, dorsal aspect of the second metatarsal head; dorsal aspect of the second metatarsal base; dorsomedial aspect of the first metatarsal head; dorsomedial aspect of the first metatarsal base; dorsolateral aspect of the fifth metatarsal base; dorsolateral aspect of the fifth metatarsal head; medial aspect of the head of the proximal phalanx of the hallux; most medial apex of the tuberosity of the navicular [30-33].”.

Page 4, lines 170 to page 5 line 172 The foot segment bases were used for the ankle, which consisted of the calcaneal markers and the heads of the first and fifth metatarsals. The angles were calculated using this base associated with the leg base.”

Page 3, lines 124-126 “All participants performed the same test order, starting with the stair descent, followed by the single-leg step down, and finalized the assessment with the single-leg squat tasks.”

Results

Are presented satisfactorily, although if the tables are formatted it would improve their readability.

Response: We appreciate your suggestion. As the other reviewers also suggest improving the results presentation, we decided to replace all the tables for figures in the main manuscript, but we kept the tables as supplementary files in case someone want to consult the exact numeric data.

Line 172 Revise this value (“compared to stair descents (0.88 Nm/kg;” ) as it is not the same in table 2

Response: Thank you for your careful revision. We apologize for that. We changed it to -0.88Nm/Kg (Page 6, line 215).

Discussion

Please discuss the step's high eventual influence on some of the results obtained.

Response: We appreciate your comment. We added the following:

Page 9, lines 308-317: “As for kinematic variables of the knee joint, a lower flexion peak at single-leg squats compared to the other two tasks corroborated the findings by Lewis et al [25]. They also observed a lower flexion peak in single-leg squats compared to single-leg step downs. It is essential to highlight that the height of the step can influence these results, since differences between single-leg squats and single-leg step downs were reported using steps with heights of 20 and 16 cm, but not when single-leg step downs were performed at a step height of 24 cm [25]. The step height to perform the single-leg step down may determine distinct trunk and lower limb strategies since the individual has to touch the ground with the heel of their non-dominant lower limb [25]. Specifically considering the knee, the greater the step height, a greater knee flexion would be necessary.”

Conclusion

Are presented satisfactorily. However, I believe the paper would beneficiate from highlighting the study's relevance to clinical practice.

Response: We sincerely appreciate your positive opinion. We added a paragraph in the discussion section about the clinical practice (Page10, lines 382-385) and the following sentence in the conclusion section as suggested:

These results could help choose which single-leg task would fit better within a specific clinical purpose. For example, the single-leg step down could be used if the goal is to expose the individual to greater articular demand.”

We once again thank you for the comments

Best regards

Authors

Reviewer 2 Report

The work is very well organized. It has a relevant objective and with strong contribution in the context of knee biomechanics among young people. Although this inference about the differences between gender was not possible, I consider the proposal interesting and I hope that future studies can answer what the authors wished to investigate.

I made small considerations in the structure of the text, but that in no way misconsiders the current version. I hope that these minimal revisions will add to what has already been developed with brilliance.

The small considerations are highlighted in the revised version (attached).

Author Response

We would like to thank you for your time spent and comments. We believe that the suggestions and queries from the review helped to improve the quality of our research report. We have followed the suggestions and highlighted all changes in yellow throughout the text. We have included a point-by-point cover letter.

The work is very well organized. It has a relevant objective and with strong contribution in the context of knee biomechanics among young people. Although this inference about the differences between gender was not possible, I consider the proposal interesting, and I hope that future studies can answer what the authors wished to investigate.

I made small considerations in the structure of the text, but that in no way misconsiders the current version. I hope that these minimal revisions will add to what has already been developed with brilliance. The small considerations are highlighted in the revised version (attached).

Response: We sincerely appreciate your positive comments. We considered all comments made and change the current version as suggested.

Comment 1: Include the type of the proposed study: a cross-sectional study.

Response: It was included as suggested. The revised version of the title is “Knee kinetics and kinematics of young asymptomatic participants during single-leg weight-bearing tasks: task and sex comparison of a cross-sectional study.”

Comment 2: Include in table 1 how many of both sexes are active and irregularly active.

Response: We included this information in the Table 1 as suggested. Thank you.

Comment 3: Complete with more information the execution of tests/tasks: individuals were shoes? was there some kind of prior movement? did the individuals come walking or was there a rest time before the tests/tasks? what kind of clothes did they wear?

Response: Thank you for your comment. We completed the information you asked at page 3, lines 117-122:

Participants wore workout clothes (tight shorts, and women also wore a top) and performed the tasks without shoes. They had no standard rest period before familiarization. However, the time to collect participant data, position the retroflective marks, and the system's calibration was a sufficient interval to rest. Then, participants performed 4 repetitions of each test (stair descent, single-leg step down, and single-leg squat tasks) to familiarize themselves with the tasks.”

Comment 4: How was the sample calculation done? It would be important to describe here or if it was not done, to signal the limitations of the study.

Response: No, we did not perform any a priori sample size calculation. We added it at the limitations statement (Page 9, lines 345-348):

“Maybe we could have distinct distributions or apply different statistical methods with larger sample sizes. It would also be desirable to perform a multivariate analysis. As we did not perform a priori sample size estimation, we might have been underpowered to detect some subtle differences between tasks or sexes”

Comment 5: Perhaps Table 2 deserved to present the data, differentiating between the sexes. To support the hypothesis pointed out and show whether biomechanical differences were found between the sexes.

Response: The information that you suggested to be on table 2 are those presented in table 3. In the first version, we opted to display the results of both main study questions separately. However, considering your suggestion and those from other reviewers, we replaced all the tabled to figures. The figures 3 and 4 of the current version present together the data of total sample and stratified by sexes.  Tables were kept as supplementary files to provide detailed numeric data.

Lines 238-244 . Comment 6: Is this true in all individuals, or was there a difference between the sexes?

Response: Thank you for your observation. We added a sentence in the end of this paragraph clarifying that these observations were related to total sample and that no substantial sex difference were observed.  

Page 8, lines 290-291:All the differences mentioned were observed only when the total sample was considered, and no substantial differences between the sexes were observed.”

Comment 7: Before the conclusions session: What would be the strength of this job? What contributions can these analyses generate for the health of the general population, especially among young people and in the recommendation of physical activity modalities (type, intensity) on a regular basis and promoting health?

Response: We appreciate your suggestion. We added the following paragraph (Page 10, lines 354-369) at the discussion section.

Strengths of this study include comparing three single-leg weight-bearing tasks, which are highly used in practice but, to the best of our knowledge, have never been biomechanically compared before. Understanding the similarities and differences among the three assessed tasks and between sexes may guide some practical and/or clinical decisions. For example, if the aim is to expose a young and asymptomatic individual to a single-leg weight-bearing tasks program, our results would guide to adequate the exercises to the objectives. Our results suggest that the progressive implementation of single leg-tasks in the rehabilitation program may begin with stair descent followed by a single-leg step down and finally the implementation of a single-leg squat when considering the articular moment. Moreover, in terms of range of motion in knee frontal plane, clinicians should be aware that the single-leg step down is the one that requires the greater range. So, if the goal is to stimulate greater knee mobility in the frontal plane, the single-leg step down would be preferred among the three assessed herein; however, if there is any clinical precaution to such mobility, this task could be avoided at this timepoint. Finally, as we did not observe any differences in the knee peak angle in the sagittal plane, all the three tasks could be equally chosen.

We once again thank you for the comments

Best regards

Authors

Reviewer 3 Report

Dear Authors,

It is my pleasure to review your study but I have a lot of doubts.

General information:

-The abstract needs correction. It should be divided into: Introduction, M&M,  

etc.

-More key words can be added.

-In title replace “subjects” by “patients” (please be humble and use patients instead of subjects).

Introduction:

Line 64 replace “subjects” by “patients”.

M&M:

-Single-leg squat is a difficult exercise, did each participant do it correctly? Did it require preparation? Even a physically fit person has a problem with carrying it out.

-Did you apply a sample size calculation?

-What was the percentage of right-legged people?

-"The Euler angle convention" - it should be explained briefly, not every reader may know it.

-The discussed angles and movements in the joints in the text are complicated. It is difficult to understand, visualize for the reader. I propose to present it additionally using graphics (figure).

Result:

-Table No 2,3 and 4 could be presented in a more readable way for the reader. A lot of data, difficult to interpret for the reader.

Discussion:

-References in the discussion is too old, it needs to be changed.

-A limitation of the present study - you can complete with: study on a larger group of patients.

Author Response

We would like to thank you for your comments. We believe that the suggestions and queries from the review helped to improve the quality of our research report. We have followed the suggestions and highlighted all changes in yellow throughout the text. We have included a point-by-point cover letter.

Dear Authors,

It is my pleasure to review your study but I have a lot of doubts.

Response: We appreciate your time spent on this revision and thank you for the comments. We revised the manuscript aiming to address all the points raised by you. Thank you for the opportunity to review it. We hope that the changes made would clarify the doubts.

General information:

-The abstract needs correction. It should be divided into: Introduction, M&M,  

etc.

Response: We understand your suggestion, however, we followed the journal´s templates that mention: “We strongly encourage authors to use the following style of structured abstracts, but without headings”.

-More key words can be added.

Response: We added four new key words “kinematic; exercise; functional assessment; biomechanics”

-In title replace “subjects” by “patients” (please be humble and use patients instead of subjects).

Response: As the participants of our study does not have a medical condition which is the interest of the investigation, we consider your suggestion but changed the term “subject” to “participant” all over the manuscript.

Introduction:

Line 64 replace “subjects” by “patients”.

Response: As explained for the title, we replaced the term subjects by participants.

M&M:

-Single-leg squat is a difficult exercise, did each participant do it correctly? Did it require preparation? Even a physically fit person has a problem with carrying it out.

Response: All the single-leg tasks were performed 4 times as a familiarization phase as described at page 3, lines120-122. Although we agree that it could be the most challenge task, we did not have to exclude any participant due to difficulty to perform the single-leg squat.  

-Did you apply a sample size calculation?

Response: No, we did not perform a priori sample size calculation because we did not have any prior data of the three tasks of interest to perform this calculation. Also, we did not make r any pilot study. The greatest consequence of an underpowered sample size in our study design is the probability of a type II error (false negative) in the results. We added it as a limitation.

Page 9, lines 342-348: “A limitation of the present study is the impossibility of verifying the interaction between the two factors (tasks and sex) due to the distribution of residual data, which could show different patterns in performing the tasks. Maybe we could have distinct distributions or apply different statistical methods with larger sample sizes. It would also be desirable to perform a multivariate analysis. As we did not perform a priori sample size estimation, we might have been underpowered to detect some subtle differences between tasks or sexes.

-What was the percentage of right-legged people?

Response: The percentage of right-legged people was 87%. We added the proportion of each lower limb dominance side in the Table 1.

-"The Euler angle convention" - it should be explained briefly, not every reader may know it.

Response: We appreciate your comment. In fact, we detect that it was incorrectly described. We used Visual 3D to analyze our data and this software uses the Cardan angles. However, taking your observation, we added the following paragraph to clarify the determination of our joint coordination system.

Page 4, line 156 to page 5, line 174: “The joint coordination system was used to analyze the hip, knee, and ankle angles [37], adopting an XYZ sequence. First, a Cartesian coordinate system is established for each of the two adjacent body segments, defined based on bony landmarks. The common origin of both systems defines the neutral position, and it is the reference for the linear translation. Secondly, the joint coordination system is established based on the two Cartesian coordinate systems. Two of the joint coordination system axes are body fixed, and one is not fixed. Finally, the joint motion, including three rotational and three translational components, is defined based on the joint coordination system [38]. Local reference systems of the thigh and leg were built to calculate the knee joint angles. Markers on the greater trochanter of the femur and the lateral and medial epicondyles of the femur were used to build the thigh segment bases. The markers on the head of the fibula, lateral malleolus, and medial malleolus were used to build the leg segment bases. The hip segment base was built with the markers on the right and left anterosuperior iliac spine and the midpoint between the markers of the two posterosuperior iliac spines. The angles were calculated using this base associated with the thigh base. The foot segment bases were used for the ankle, which consisted of the calcaneal markers and the heads of the first and fifth metatarsals. The angles were calculated using this base associated with the leg base. The definitions of the joint coordinate system recommended by the International Society of Biomechanics [38] were used.”

-The discussed angles and movements in the joints in the text are complicated. It is difficult to understand, visualize for the reader. I propose to present it additionally using graphics (figure).

Response: Thank you for your suggestion. We added a Figure to represent the movements and signal definitions we used in our manuscript (Please, see figure 2).

-Table No 2,3 and 4 could be presented in a more readable way for the reader. A lot of data, difficult to interpret for the reader.

Response: We appreciate your suggestion. As the other reviewers also suggest improving the results presentation, we decided to replace all the tables for figures in the main manuscript, but we kept the tables as supplementary files in case someone want to consult the exact numeric data.

Discussion:

-References in the discussion is too old, it needs to be changed.

Response: We appreciate your suggestion. We added 12 more recent references (References 16-19 and 39-46) in this section and kept those that was directly related to our data.

-A limitation of the present study - you can complete with: study on a larger group of patients.

Response: Thank you for the suggestion. We added it at page 9, lines 345-348: “Maybe we could have distinct distributions or apply different statistical methods with larger sample sizes. It would also be desirable to perform a multivariate analysis. As we did not perform a priori sample size estimation, we might have been underpowered to detect some subtle differences between tasks or sexes”

We once again thank you for the comments

Best regards

Authors

Reviewer 4 Report

The authors want to comparate the differences among sex  in the Knee kinetics and kinematics of young asymptomatic subjects during single-leg weight-bearing tasks. The paper focus its attention in the different neuromuscular pattern during a single-leg task. The soundness is good in terms of prevention of injuries and rehabilitation approach. Nevertheless, there are some critical issues to be addressed:

Introduction: Have you noticed in literature some correlation between injuries and keee kinetic and kinematic characteristics? Moreover, have you explication about the neuromuscular control of knee movement?

Please, cite : De Sire A. et al. Neuromuscular Impairment of Knee Stabilizer Muscles in a COVID-19 Cluster of Female Volleyball Players: Which Role for Rehabilitation in the Post-COVID-19 Return-to-Play?, 2022, Applied sciences

Table: please, replace the term "mass" with "height"

Materials and Methods: Why have you deleted the very active and sedentary from analysis? Are there some information about?

Have you adjusted the data with a multivariate analysis (an explication is in the introduction? Please, specify this point and list the variables.

Results: It would be desirable to depict a table with the baseline characteristics and differences between male and female

Discussion: The authors have underlined that the potential confounding factor "Height" was normalized. This part should be added in statistical analysis.

Taken together, the paper is well written and suitable for revision

Best Regards

Author Response

We would like to thank you for your comments. We believe that the suggestions and queries from the review helped to improve the quality of our research report. We have followed the suggestions and highlighted all changes in yellow throughout the text. We have included a point-by-point cover letter.

The authors want to comparate the differences among sex  in the Knee kinetics and kinematics of young asymptomatic subjects during single-leg weight-bearing tasks. The paper focus its attention in the different neuromuscular pattern during a single-leg task. The soundness is good in terms of prevention of injuries and rehabilitation approach. Nevertheless, there are some critical issues to be addressed:

Response: We sincerely appreciate your positive comments. We considered all comments made and change the current version as suggested.

Introduction: Have you noticed in literature some correlation between injuries and keee kinetic and kinematic characteristics? Moreover, have you explication about the neuromuscular control of knee movement?

Please, cite : De Sire A. et al. Neuromuscular Impairment of Knee Stabilizer Muscles in a COVID-19 Cluster of Female Volleyball Players: Which Role for Rehabilitation in the Post-COVID-19 Return-to-Play?, 2022, Applied sciences.

Response: We appreciate your observation. There is no evidence that kinetic and kinematics aspects are risk factors to knee injuries. We revised the introduction and clearly state it (Page 1, line 41 to page 2 line 47). We also added in the discussion section the paper you asked us to cite (Page 10, lines 370-377).

Page 1, line 41 to page 2 line 47 “Current evidence indicates that kinematics and kinetic aspects are not risk factors for knee injuries [16-18], except when they are intrinsically related to the task, such as running-related injuries [19]. Still, these biomechanical differences may be considered when planning the exercise protocols. Depending on the rehabilitation's objective or phase (early or advanced), some patients might have therapeutic restrictions considering the articular range of motion or benefit from a progressive demand according to the rehabilitation's objective or phase (early or advanced).”

Page 10, lines 370-377 “Several factors can contribute to altered kinetics and kinematics during single-leg weight-bearing tasks, such as active or passive stiffness, altered neuromuscular control, or muscle strength [40-43]. It should be highlighted that, currently, there is no evidence that altered and/or distinct kinetic and kinematic parameters are risk factors for knee-related injuries or painful conditions [16-18]. Nevertheless, altered biomechanics can be observed in patients with anterior cruciate ligament repair, knee osteoarthritis, and patellofemoral pain [44-46]. However, as we did not assess any pathology or painful knee condition, the practical implications mentioned before may not be the same for these cases.”

Table: please, replace the term "mass" with "height"

Response: We replaced it by weight.

Materials and Methods: Why have you deleted the very active and sedentary from analysis? Are there some information about?

Response: Because we decide to maintain a more homogeneous sample. We already had two factors to consider in our project (task and sex). At the time, we decided that both very active and sedentary could contribute to greater variability of the task’s execution due to the influence of task ability, strength and neuromuscular control. However, we considered your observation and added it as suggestion to future studies as can be observed at page 10, lines 351-353:

“Another perspective for future studies is the inclusion of sedentary and very active participants, considering the activity level as a factor that could be associated with the kinetic and kinematic parameters.”

Have you adjusted the data with a multivariate analysis (an explication is in the introduction? Please, specify this point and list the variables.

Response: No, we applied non-parametric tests to analyze our data as we did not meet some of the assumptions needed to the parametric tests. It is the main reason we were not able to add any co-variable neither to perform multivariate analysis. In the current version, we mention it in the limitation’s statements.

Page 9, lines 342-346 “A limitation of the present study is the impossibility of verifying the interaction between the two factors (tasks and sex) due to the distribution of residual data, which could show different patterns in performing the tasks. Maybe we could have distinct distributions or apply different statistical methods with larger sample sizes. It would also be desirable to perform a multivariate analysis.”

Results: It would be desirable to depict a table with the baseline characteristics and differences between male and female

Response: We presented baseline characteristics in Table 1 (Page 3) and mentioned it in the text.

(Page 2, lines 88-90)“The final sample included 30 individuals (15 males and 15 females), and their characteristics are described in Table 1. There were significant differences between sexes in the height and weight of the participants (p<0.05)..”

Discussion: The authors have underlined that the potential confounding factor "Height" was normalized. This part should be added in statistical analysis.

Response: We appreciate your comment. We meant it was normalized in the data processing, not in analysis because the internal joint moment was normalized by the mass and height of the participants. This information was described at the methods section (page 4, lines 153-155):The internal joint moments of the ankle, knee and hip in the frontal and sagittal planes were calculated using inverse dynamics and normalized by the weight and height of the participants.” However, in the current version, we revised the discussion to clarify it, as can be observed at page 9, lines 341-342: “However, the normalization of data by height and weight in this study eliminates this influence in the joint moment.”

Taken together, the paper is well written and suitable for revision

Best Regards

Response: we sincerely appreciate your kind comment.

We once again thank you for the comments

Best regards

Authors

Round 2

Reviewer 3 Report

Dear Authors,

Thank you so much for changes.

Currently, I have no objections. The article presents much better.

General information:

-The abstract needs correction. It should be divided into: Introduction, M&M OK

-More key words can be added. OK

-In title replace “subjects” by “patients” (please be humble and use patients instead of subjects). OK

Introduction:

Line 64 replace “subjects” by “patients”. -Single-leg squat is a difficult exercise, did each participant do it correctly? Did it require preparation? Even a physically fit person has a problem with carrying it out. OK

-Did you apply a sample size calculation? OK

-What was the percentage of right-legged people? OK

-"The Euler angle convention" - it should be explained briefly, not every reader may know it. OK

-The discussed angles and movements in the joints in the text are complicated. It is difficult to understand, visualize for the reader. I propose to present it additionally using graphics (figure). OK

-Table No 2,3 and 4 could be presented in a more readable way for the reader. A lot of data, difficult to interpret for the reader. OK

Discussion:

-References in the discussion is too old, it needs to be changed. OK

-A limitation of the present study - you can complete with: study on a larger group of patients. OK

In my opinion, the article can be published in IJERPH.

Reviewer 4 Report

I would like to thank the authors for the robust review of the paper. At the light of the revision performed by authors, the paper is suitable for fully publication

best regards